# Research on multi-agent genetic algorithm based on tabu search for the job shop scheduling problem

Chong Peng[1]*, Guanglin Wu[1], T. Warren Liao[2], Hedong Wang[1]

**1** School of Mechanical Engineering and Automation, Beihang University, Beijing, China, **2** Department of Mechanical and Industrial Engineering, Louisiana State University, Baton Rouge, LA, United States of America

* pch@buaa.edu.cn

**Data Availability Statement:** All relevant data are within the manuscript and its Supporting Information files.

**Funding:** The research was supported by National Natural Science Foundation of China (Grant No. 51875029) and was supported by the National

## Abstract

The solution to the job shop scheduling problem (JSSP) is of great significance for improving resource utilization and production efficiency of enterprises. In this paper, in view of its non-deterministic polynomial properties, a multi-agent genetic algorithm based on tabu search (MAGATS) is proposed to solve JSSPs under makespan constraints. Firstly, a multi-agent genetic algorithm (MAGA) is proposed. During the process, a multi-agent grid environment is constructed based on characteristics of multi-agent systems and genetic algorithm (GA), and a corresponding neighbor interaction operator, a mutation operator based on neighborhood structure and a self-learning operator are designed. Then, combining tabu search algorithm with a MAGA, the algorithm MAGATS are presented. Finally, 43 benchmark instances are tested with the new algorithm. Compared with four other algorithms, the optimization performance of it is analyzed based on obtained test results. Effectiveness of the new algorithm is verified by analysis results.

## 1 Introduction

Under current rapid development of social economy, whether an enterprise can quickly respond to market demands with limited resources under the premise of ensuring high efficiency, high quality, high output and low cost largely determines the development and destiny of it. In a production process of products, an effective solution of the JSSP is an extremely important link in achieving the above goals. Job shop scheduling means optimization of product manufacturing time or manufacturing cost is satisfied as far as possible by reasonably arranging processing order of each job to be processed on each machine based on existing machine resources and job raw materials under the condition of satisfying realistic constraints (product delivery date, product production process route and available resources).

The importance of JSSPs attracts many scholars to conduct research in this field. Garey and Sethi proved that JSSPs had non-deterministic polynomial (NP) characteristics [1]. For this feature, methods for solving JSSPs can be divided into two categories: exact methods and approximation methods. Exact methods include mathematical programming method, branch

Science and Technology Major Project "High-Grade CNC Machine Tools and Basic Manufacturing Equipments" (Grant No. 2016ZX04004006) to CP.

**Competing interests:** The authors have declared that no competing interests exist.

and bound method, and so on [2,3]. However, exact methods mentioned above are only applicable to small-scale JSSPs. As complexity of JSSPs increases, applicability of exact methods decreases continuously. Approximation methods include priority dispatching rules, shifting bottleneck procedure, meta-heuristic algorithms, and the like. At present, meta-heuristic algorithms such as GA, ant colony algorithm (ACO), particle swarm optimization algorithm (PSO), simulated annealing algorithm (SA), neural network, tabu search (TS) and artificial bee colony algorithm (ABC) have been widely used in various field and have shown good performance [4–6].

GA was presented by Holland [7] and was extensively applied to solving scheduling problems. Its excellent searching performance is favored by many researchers in this field. However, application of GA in dealing with complex JSSPs is limited due to its premature and local convergence. In response to these shortcomings, researchers proposed a number of improvement measures, such as two-stage genetic algorithm, island model genetic algorithm, hybrid genetic algorithm , and so on. Xu and Li [8] proposed immune genetic algorithm by combining immune theory with GA, which improved the global search performance of GA. Kurdi [9] proposed a new island model genetic algorithm(NIMGA), which contains a new naturally inspired evolution model and a new naturally inspired migration selection mechanism, to improve effectiveness of classical island model genetic algorithm. The new algorithm improved diversification of the search and delayed premature of GA. Chang and Liu [10] proposed a hybrid genetic algorithm by using the Taguchi method to optimize the parameters of a GA. Robustness and good optimization performance of the new algorithm were verified after applying to solve some distributed and flexible job-shop scheduling problems. Moreover, multi-agent systems have gained more and more applications in production scheduling due to their flexibility and adaptability to open and dynamic real-world environments and synergy mechanism between multiple agents. Liu et al. [11] designed a multi-agent-based solution system and successfully solved the complex 7000 queen problem. Chen et al. [12] proposed a hybrid flow shop rescheduling algorithm for perishable manufacturing systems. Products with different deadlines, values, due dates and stochastically failed operational units in the system were simulated by agents. The product agents can search the optimal scheduling path with the new algorithm.

A MAGATS is proposed for solving JSSPs in this paper. To improve global search ability of the algorithm and maintain diversity of the population, on the basis of establishing a multi-agent grid environment based on GA, this paper designs a neighbor interaction operator, a mutation operator based on neighborhood structure and a self-learning operator. The algorithm combines high efficiency and simple operation of GA, synergy mechanism of a multi-agent system and global search ability of TS, which can make up for premature and local convergence of GA to some extent and improve search efficiencies and optimization ability of the algorithm. Then, JSSPs can be solved more efficiently and accurately. Resource utilization of enterprises can be promoted. Market demands can be faster respond to, which is of great significance to development of enterprises and a country.

The remaining of this paper is organized as follows. The model of the JSSP and the process of achieving a MAGA are illustrated in Section 2. Section 3 introduces the establishment of MAGATS. In Section 4, 11 benchmark instances are selected to verify optimization performance of MAGATS compared with GA and MAGA. 43 benchmark instances are used to test optimization performance of the new algorithm. Based on test results and comparison with other algorithms, optimization performance of MAGATS is studied. The conclusion is made in Section 5.

## 2 The achievement of MAGA

This section firstly introduces the model of the JSSP to be solved. Then combining multi-agent synergy theories and GA, a MAGA is proposed.

### 2.1 Model of the JSSP

A JSSP is usually defined as follows.

$n$ jobs are machined on $m$ machines, respectively denoted as sets $J = \{J_i|i = 1, 2, \ldots, n\}$, $M = \{M_i|i = 1, 2, \ldots, m\}$, where $J_i$ is a job code, $M_i$ is a machine code. Each job has a specific process. Orders that jobs use machines are assigned. The processing time required for each operation of each job on the corresponding machine is also given. The specific content of scheduling is to determine processing sequences of jobs on each machine and the starting time of each job, which makes makespan (recorded as $C_{max}$) shortest. Besides that,

(1) Sequences of different jobs on a same machine have no constraints;

(2) Once a job is started on a machine, it is completed until its process is completed. Each machine can only process one job at the same time, and it is assumed that no machine failed;

(3) All jobs arrive at the same time and only flow through each machine once.

It is assumed that the total number of operations for all jobs is Q, G = $\{1, 2, \ldots, Q\}$ is named as an operation set; $J_i$ represents the job subordinate to operation $i$; Machines required for processing operation $i$ are marked as $M_i$; The starting time of operation $i$ is denoted by $S_i$; Time needed for processing operation $i$ is recorded as $P_i$; The sequence of two adjacent operations of a job is represented by $\rightarrow$. Take $i \rightarrow j$ for example, operations $i, j$ belong to one job, that is, $J_i = J_j$ and operation $i$ is before operation $j$. So, the model of the JSSP can be described as follows.

$$\min\{\max S_i + P_i\} \tag{1}$$

$$
\begin{aligned}
s.t. : \quad & S_i \geq 0, \ \forall i \in G \\
& S_j \geq S_i + P_i, i \rightarrow j, \forall i, j \in G \\
& (S_j \geq S_i + P_i) \vee (S_i \geq S_j + P_j) \forall i, j \in G, i \neq j, M_i = M_j
\end{aligned}
\tag{2}
$$

In Eq (2), the second constraint represents an operation sequence constraint, that is, the start time of any one operation of a job must be later than the completion time of its previous operation. The third constraint represents a resource constraint of machines, that is, each machine can only process one job at the same time.

### 2.2 Construction of MAGA

By simulating the "survival of the fittest" rule in nature, classic GA calculates fitness values of all chromosomes in each evolution process at first, and then preferentially selects individuals with high fitness in a certain way, and weeds out ones with low fitness with a high probability. But in nature, things often evolve locally and then expands globally, which is like co-evolution of multi-agents in a multi-agent system. A multi-agent system is a distributed system composed of multiple agents. Each agent can be either hardware or software. It can sense and respond to local environment and communicate with neighbor agents to complete complex tasks. In a multi-agent system, co-evolution of multi-agents in a local environment can better reflect processes of evolution in nature. Therefore, GA is firstly improved to some extent by combining GA with multi-agent synergy theories.

**2.2.1 Multi-agent grid environment based on GA.** GA is a random search algorithm formed by simulating evolution processes of biology. The optimization process is realized by three basic operators: selection-reproduction, crossover and mutation. As shown in Fig 1, GA

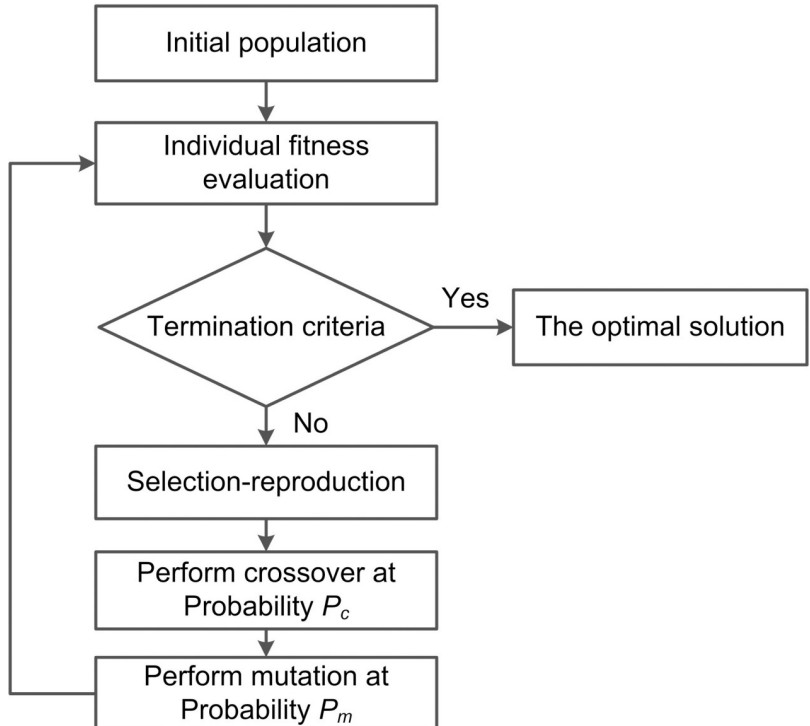

**Fig 1. Flow chart of GA.**

consists of five basic elements: encoding and decoding, design of initialing population, fitness evaluation function design, design of genetic operations such as selection-reproduction, crossover, mutation, and setting of genetic parameters.

Based on GA, a multi-agent grid environment was built. In this environment, each chromosome in GA is treated as an independent agent. Firstly, a neighbor interaction operator was designed to realize the optimization function of multi-agent systems. Secondly, to maintain diversity of population, a mutation operator based on neighborhood structure and a self-learning operator were designed and introduced. The optimization algorithm based on the multi-agent grid environment is called a MAGA. The overall flow chart is shown in Fig 2.

Compared Fig 1 with Fig 2, the process of MAGA is basically the same as that of GA. Similar to GA, MAGA has the same process of encoding and decoding each agent in the grid environment.

**2.2.1.1 Encoding and decoding of GA.** Encoding is a gene representation of solutions of JSSPs. It is the primary problem faced by GA for optimization of JSSPs, which plays a key role in improving optimization effectiveness of the algorithm. Encoding methods of GA currently used for JSSPs can be summarized into two categories: direct encoding and indirect encoding. Direct encoding methods include operation-based encoding, job-based encoding, job pairs-based encoding, completion time-based encoding, random key-based encoding, while indirect encoding methods usually include precedence table-based encoding, priority rules-based encoding, disjunctive graph-based encoding and machine-based encoding [13–16]. Combined with advantages and disadvantages of various encoding methods, the operation-based encoding is more prominent than other encoding methods, and it is the most widely used method to solve JSSPs. Adopting operation-based encoding, Arbitrary arrangement of jobs can be expressed as feasible scheduling. The corresponding decoding scheme is simple and easy to operate. Moreover, feasible scheduling can always be obtained after replacing chromosomes.

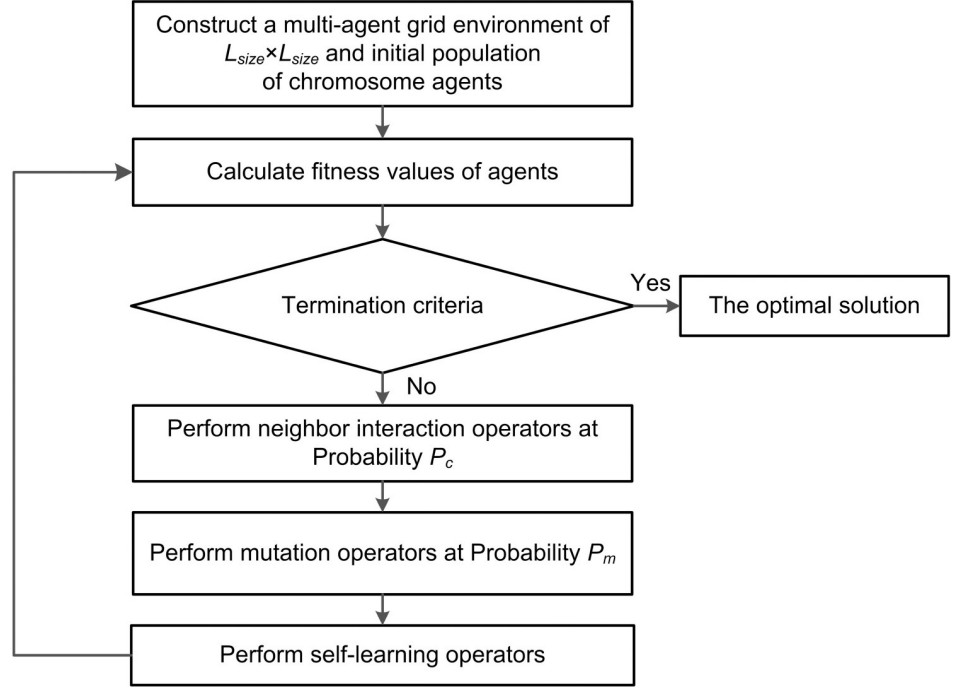

**Fig 2. Flow chart of MAGA.**

Although this method only has a half-Lamarkian characteristic (Lamarkian characteristic refers to the ability of inheriting good information from parent chromosomes), performance of the algorithm can be enhanced by improving design of genetic operations, thereby improving its genetic characteristics. In summary, the operation-based encoding method is used to encode JSSPs. In this method, each chromosome is represented by a gene sequence. Each gene sequence contains $n \times m$ ($n$ jobs, $m$ machines) genes representing operations, which is an arrangement of all operations.

Existing decoding methods include semi-active decoding and active decoding. Active decoding can make decoded operations more concentrated, which can improve search efficiencies of an algorithm [17]. Therefore, an insertion strategy is adopted to decode chromosomes. Every operation in a sequence is sequentially arranged on a corresponding machine. When constraints are satisfied, every operation is started as early as possible. Finally, an active chromosome of the JSSP, that is, an active scheduling solution, is obtained. To illustrate the decoding process, data given in Table 1 are taken as an example. Processing time "4-5-11-3" in the second row of Table 1 indicates processing time required for operations of job $J_1$ on the machines $M_4$, $M_3$, $M_1$, $M_2$, respectively.

The process of decoding a chromosome is shown in Fig 3. For example, an operation sequence [1 3 1 4 2 3 2 4 3 3 4 1 4 1 2 2] generated based on data of Table 1 is actively decoded.

**Table 1. 4×4 JSSP.**

| Job | Machine | Processing time |
|---|---|---|
| $J_1$ | $M_4$-$M_3$-$M_1$-$M_2$ | 4-5-11-3 |
| $J_2$ | $M_3$-$M_2$-$M_1$-$M_4$ | 5-2-5-1 |
| $J_3$ | $M_3$-$M_4$-$M_2$-$M_1$ | 2-5-9-3 |
| $J_4$ | $M_2$-$M_3$-$M_4$-$M_1$ | 6-2-4-5 |

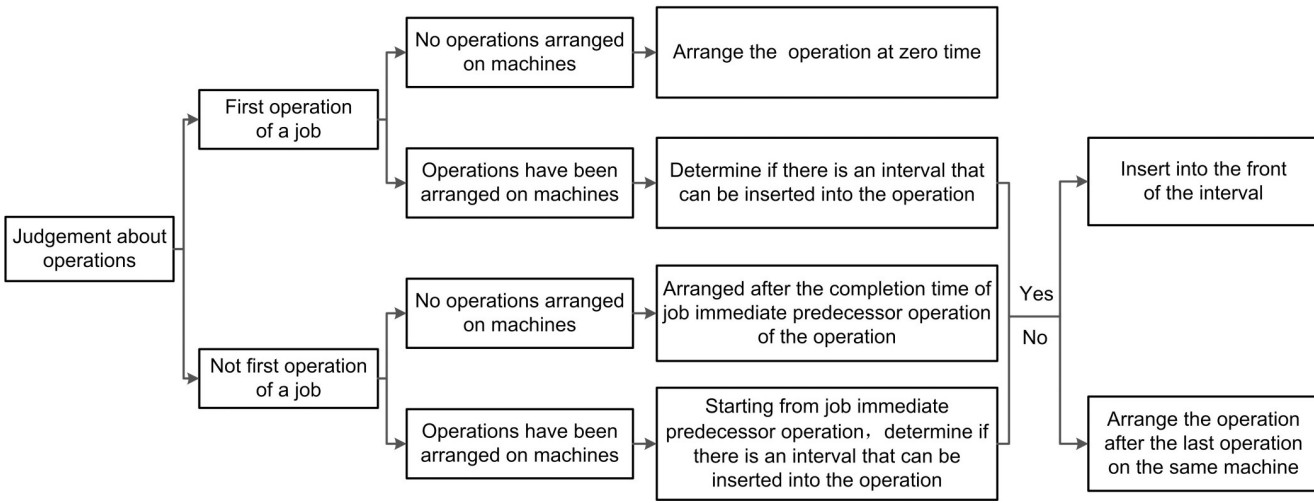

**Fig 3. Active decoding of JSSP.**

When the 3rd operation of job $J_1$ is to be arranged, that is, when the 3rd "1" of the operation sequence is taken, the corresponding machine $M_1$ has been arranged job $J_3$, it could be inserted in front of job $J_3$. Similarly, the 4th operation $O_{44}$ of job $J_4$ could also be inserted in front of job $J_3$, and finally $C_{max}$ could be decoded to be 34. The active scheduling solution obtained by actively decoding the chromosome is shown in Fig 4 where $O_{ij}$ represents the $j$th operation of job $J_i$. After active decoding the example chromosome, an active scheduling solution with more concentrated operations [1 3 4 1 3 2 1 2 4 3 4 4 3 1 2 2] is obtained.

**2.2.1.2 Neighbor environment and action criteria of chromosome agents.** MAGA uses encoding and decoding methods in GA to initial population and evaluate fitness of chromosome agents in the grid environment. To utilize multiple agents for collaborative optimization, three issues should be identified first.

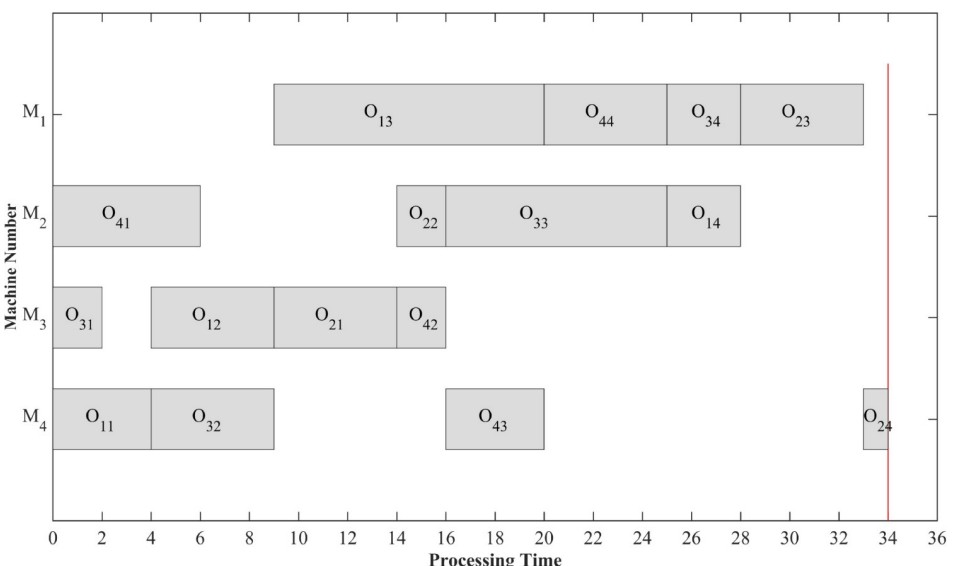

**Fig 4. Active scheduling solution.**

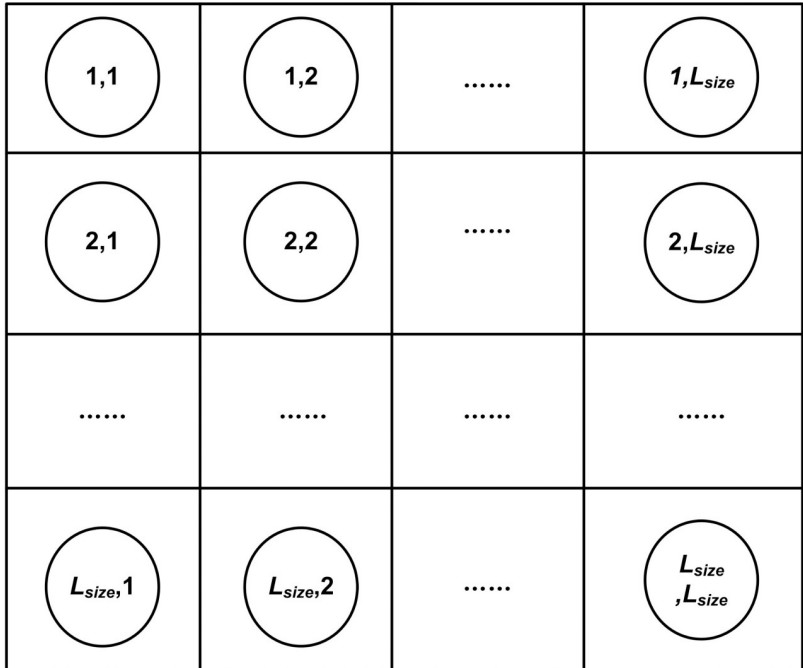

**Fig 5. Multi-agent gird environment.**

(1) Action intention definition of all agents. When using multiple agents for collaborative optimization, each agent represents a feasible solution of the JSSP. Therefore, its action intention is defined: under the premise of satisfying constraint conditions, encoding sequences are adjusted to the optimal solution in combination with its own environment.

(2) Neighbor environment definition of chromosome agents. Regard each chromosome as a separate agent, and then fix them in a $L_{size} \times L_{size}$ grid environment (as shown in Fig 5), thus forming a neighbor environment for each chromosome agent. It should be noted that the number of neighbors per chromosome is not fixed. It can be 4, 8 or more, or it can be dynamic. Since the main purpose of this paper is to combine a multi-agent grid environment with GA to study its optimization effectiveness, the number of neighbors is set to 4. It is defined as follows.

Suppose $A_{ij}$ represents a chromosome agent with coordinates $(i, j)$ in the grid environment, then a neighbor set $N_{ij}$ of it can be defined:

$$N_{ij} = \{A_{i'j}, A_{ij'}, A_{i''j}, A_{ij''}\}$$

$$i' = \begin{cases} i - 1. i \neq 1 \\ L_{size}, i = 1 \end{cases}, \quad i'' = \begin{cases} i + 1, i \neq L_{size} \\ 1, \quad i = L_{size} \end{cases}$$

$$j' = \begin{cases} j - 1. j \neq 1 \\ L_{size}, j = 1 \end{cases}, \quad j'' = \begin{cases} j + 1, j \neq L_{size} \\ j, \quad j = L_{size} \end{cases} \tag{3}$$

where $L_{size}$ represents the dimension of the grid environment. Four neighbors corresponding to each agent can be found according to Eq (3). For example, neighbors coordinates of agent $A_{22}$ are (1, 2), (2, 1), (2, 3) and (3, 2). Neighbors coordinates of agent $A_{11}$ are $(L_{size}, 1)$, $(1, L_{size})$, (2, 1) and (1, 2).

(3) Action criteria of chromosome agents. Each agent interacts with only four neighbors according to its own environment, and improves its own fitness by competing and cooperating with the best neighbors around.

**2.2.2 Design of neighbor interaction operators.** In general, design of a crossover operator should be able to inherit good features of parent chromosomes on the basis of ensuring generation of feasible solutions, so that the algorithm evolves toward the optimal solution. For JSSPs, if the scheduling solution obtained by processing two adjacent jobs $(J_i, J_k)$ on a machine is better, then inheritance of this processing order $(J_i, J_k)$ to offspring can still retain this advantage. Many scholars have proposed many crossover operators. The more successful ones are Linear Order Crossover (LOX), Precedence Operation Crossover (POX), Precedence Preservation Crossover (PPX), etc [18–20]. The POX operator proposed by Zhang et al. can better inherit superior features of parent chromosomes compared with other crossover operators [21]. Its specific implementation is shown in Fig 6.

Specific steps of POX are as follows.

(1) A job set is randomly divided into two non-empty subsets JS1, JS2, as shown in Fig 6, JS1 = {1, 3}, JS2 = {2, 4};

(2) Genes belonging to JS1 in parent1 are directly copied to child1, and genes belonging to JS1 in parent2 are directly copied into child2.Positions of the genes are retained;

(3) The remaining genes in parent2 are copied into child1, and the remaining genes in parent1 are copied into child2. Orders of the genes are preserved.

Individuals involved in the crossover in GA are from random pairing, but individuals participating in neighbor interaction only exist in the local environment of each agent. The design criteria for a neighbor interaction operator are as follows.

(1) If a current agent's fitness value is higher than its four neighbor agents, it will be retained as a winner, and its encoding sequence remains unchanged;

(2) If a current agent's fitness value is lower than the fitness value of the optimal neighbor agent, it will be replaced by the optimal neighbor. The replacement process adopts a POX operator. Unlike traditional crossover operations, each interaction only generates one offspring, that is, only child1 is retained.

There are two strategies for the replacement mechanism.

(1) Take the optimal neighbor agent as parent1 and the current agent is regarded as parent2. Obtain an optimal offspring through $\lambda$ POX operations;

(2) Take the current agent as parent1 and the optimal neighbor agent is regarded as parent2. Obtain an optimal offspring through $\lambda$ POX operations.

Strategy (1) is accepted at probability $P_o$, the other one is adopted at probability 1-$P_o$. The first strategy focuses more on concentration of optimization and is beneficial to speed up an algorithm's convergence rate. The second strategy focuses more on dispersion of optimization.

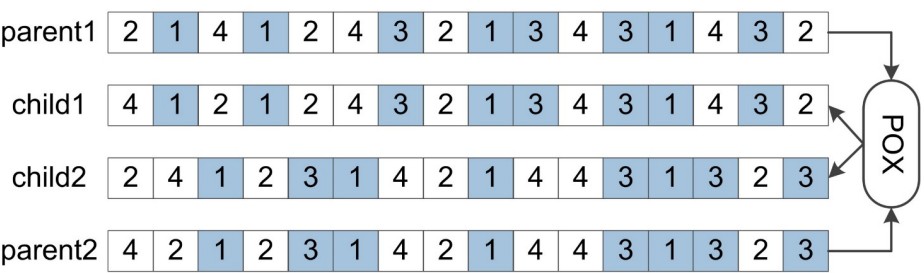

**Fig 6. Principles of POX.**

**2.2.3 Design of mutation operators based on neighborhood structure.** Mutation operators are designed to give agents a small disturbance to ensure the diversity of population. After solving the JSSP through the neighbor interaction operator, if the optimal agent obtained has high fitness, how to further improve its fitness decides whether an algorithm can find a better solution. Mutation operations commonly used in existing researches include exchange mutation, insertion mutation, reverse mutation, replacement mutation, and the like. There are two shortcomings in these mutation operations. Firstly, directions of mutation have a large blindness, and it is difficult to direct an algorithm to the global optimal solution. Secondly, these mutation operations have a greedy nature to some extent, which is easy to direct an algorithm to local optimum. A neighborhood structure is a mechanism that applies a small perturbation to a given solution to obtain another solution. Therefore, a new mutation operator named mutation operator based on neighborhood structure is presented.

**2.2.3.1 The neighborhood structure of the JSSP.** In the field of combinatorial optimization, studying the neighborhood structure of a problem is one of the most important ways to optimize solutions of the problem. Van Laarhoven et al. [22] proposed N1 neighborhood structure. New solutions were generated by randomly exchanging two adjacent operations on a critical path. It is proved that the optimal solution can be found through a limited number of interchange operations under N1 neighborhood structure. To further promote an algorithm's quality and efficiency, many researchers subsequently proposed N4, N5, N6 neighborhood structures [23–25]. Matsuo et al. found that only when the job immediate predecessor operation (JIPO) of operation $u$ or the job immediate successor operation (JISO) of operation $v$ was contained in a critical path including operations $u$ and $v$ (assuming operation $u$ is processed before operation $v$), exchanging key operations $u$ and $v$ made it possible to reduce makespan of a given solution [26]. Based on this theory, N7 neighborhood structure was proposed by Zhang et al. [27]. This new neighborhood structure can explore a wider solution space on the basis of ensuring generation of feasible solutions, so that higher quality feasible solutions can be obtained. Before introducing N7 neighborhood structure, some concepts need to be explained.

Definition 2.1 In the JSSP, each key operation $u$ has two immediate predecessor operations and two immediate successor operations, which includes: a JIPO and a JISO. The previous and next operation of operation $u$ that belong to the same job as operation $u$ are represented by $J_{pre}[u]$ and $J_{suc}[u]$ respectively. A machine immediate predecessor operation (MIPO) and a machine immediate successor operation (MISO). The previous and next operation of operation $u$ that belong to the same machine as operation $u$ are represented by $M_{pre}[u]$ and $M_{suc}[u]$ respectively.

Definition 2.2 A critical path is the longest path from the starting point 0 to the ending point 17 in a directed graph as shown in Fig 7. The path length is makespan of a scheduling solution. Fig 7 is a disjunctive graph model which took 4×4 JSSP in Table 1 as an example and used the introduction of a disjunctive graph in [14] as a basis.

Definition 2.3 Each operation on a critical path is called a key operation. As shown in Fig 8, according to the active scheduling solution in Fig 4, select a certain direction for bidirectional arcs in the disjunctive graph model, thereby forming a directed graph representation of 4x4 JSSP. At the same time, adjacent key operations group of a largest sequence processed on the same machine is called a block, such as [$O_{34}$, $O_{23}$] is a block.

Based on definitions 2.1 to 2.3, under the premise that operation $x$ and $y$ are both on the same critical path, N7 neighborhood structure can be defined as follows.

(1) When $y$ is a block tail operation and its JISO is in the critical path, operation $x$ can be moved after operation $y$, and vice versa. As shown in Fig 9.

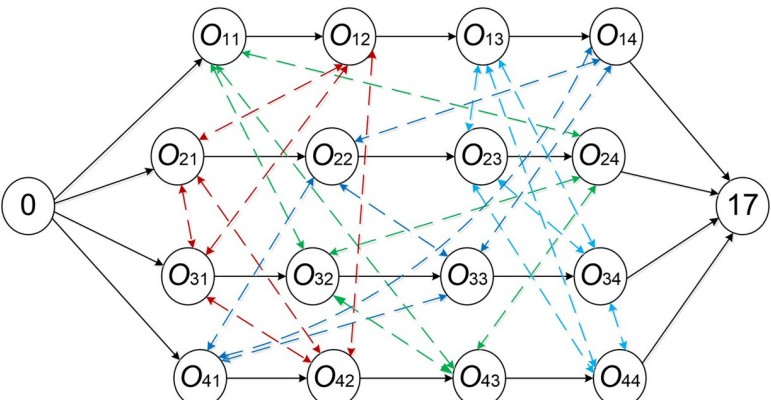

**Fig 7. A disjunctive graph model of 4×4 JSSP.**

(2) When *x* is a block head operation and its JIPO is in the critical path, operation *y* can be moved before operation *x*, and vice versa. As shown in Fig 10.

Figs 9 and 10 are key operation blocks in two critical paths including operation *x* and operation *y*. $u_1$, $u_k$ and "..." in the rectangular frame indicate intermediate operations in the block.

**2.2.3.2 Key operations search based on operation-based encoding.** Based on operation-based encoding, the first step in generating a neighborhood solution is to search for key operations. Searching for key operations can be done by a machine Gantt chart obtained after active decoding. Starting from the last completed operation, operations whose completion time immediately follows the start time of the current operation are marked as key operations. If MIPO and JIPO of an operation are encountered at the same time, the JIPO is selected as a key operation.

Take the active scheduling solution [1 3 4 1 3 2 1 2 4 3 4 4 3 1 2 2] obtained before for example, its key operation blocks are shown in Fig 11.

**2.2.3.3 Mutation mechanism based on neighborhood solutions.** According to discussions of two parts above, after obtaining key operation blocks of an agent encoding sequence, a mutation operator based on neighborhood structure is implemented as follows.

(1) Record all operation pairs that can be exchanged according to the definition of N7 neighborhood structure.

(2) Randomly select an exchangeable operation pair at probability $P_m$ to perform mutation operation.

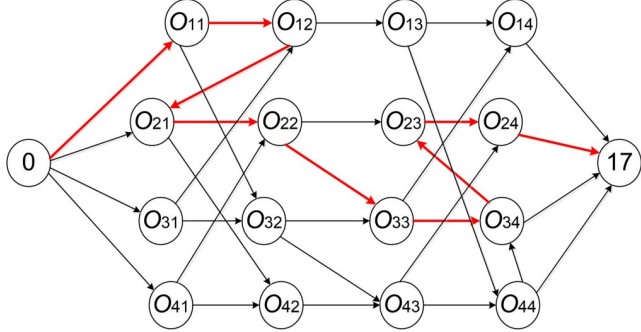

**Fig 8. An acyclic directed graph and a critical path of 4×4 JSSP.**

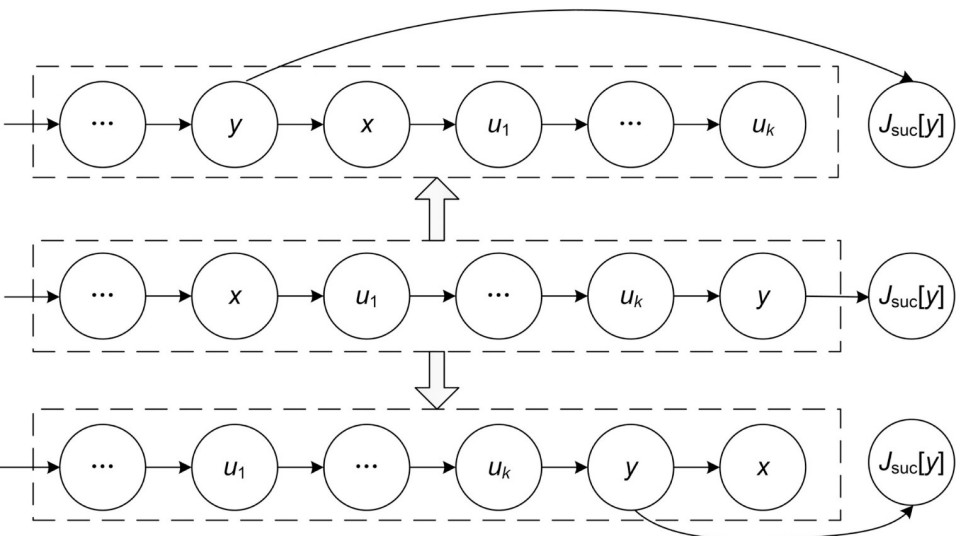

**Fig 9. JISO of a block tail operation is in the critical path.**

**2.2.4 Design of self-learning operators.** Each agent can gradually improve its fitness value through interaction operations with neighbors and mutation operations based on neighborhood structure. Similarly, take the optimal agents in each generation to construct a small grid environment with a size of $sL_{size} \times sL_{size}$. Perform $S_{gen}$ neighbor interactions and mutation operations based on neighborhood structure to obtain a new optimal agent. The flow chart is shown in Fig 12.

# 3 The achievement of MAGATS

The algorithm framework of MAGA is based on that of GA. Therefore, it is inevitable to fall into local optimal solutions when solving JSSPs. Currently, a hybrid intelligent algorithm through combining a swarm intelligence algorithm with a local search algorithm is a more

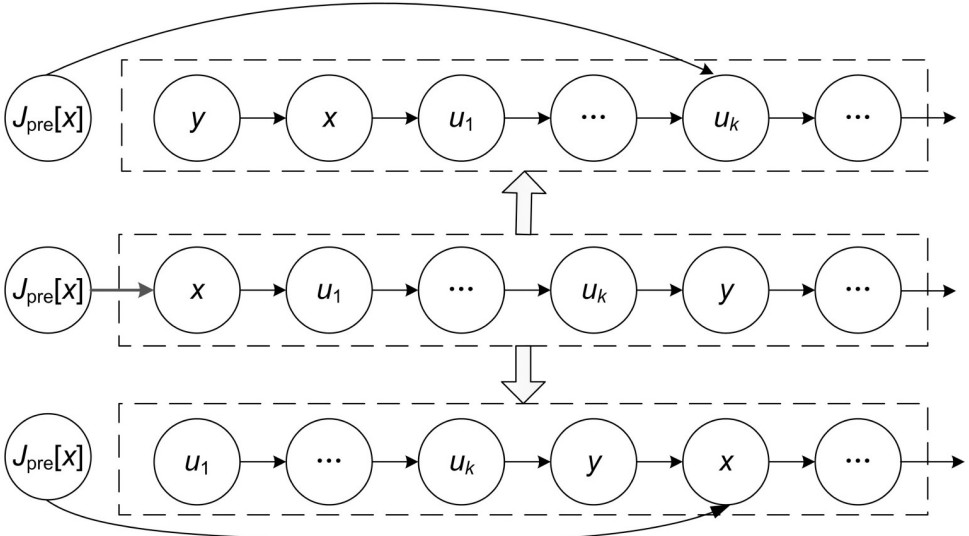

**Fig 10. JIPO of a block head operation is in the critical path.**

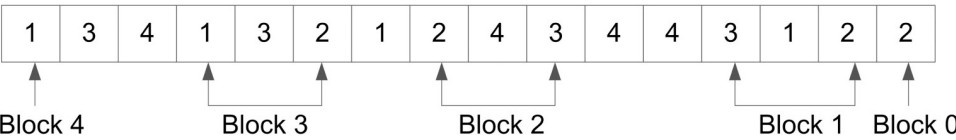

**Fig 11. Key operation blocks.**

advanced algorithm for solving JSSPs [19]. Compared with only adopting swarm intelligence algorithms, its optimization effects have been significantly improved. Currently, local search algorithms are mainly represented by TS and SA. Since SA avoids local optimization by accepting inferior solutions with a certain probability, it may return to the previous solution in the process of searching for the optimal solution, which leads to the algorithm to oscillate around a local optimal solution, thus wasting a lot of computing time. For this, on the basis of MAGA, a MAGATS is proposed by introducing TS to enhance the global search ability. The flow chart of MAGATS is shown in Fig 13.

Specific steps of MAGATS are summarized as follows.

(1) After conducting a MAGA , the best agent is set as the current solution and the optimal solution.

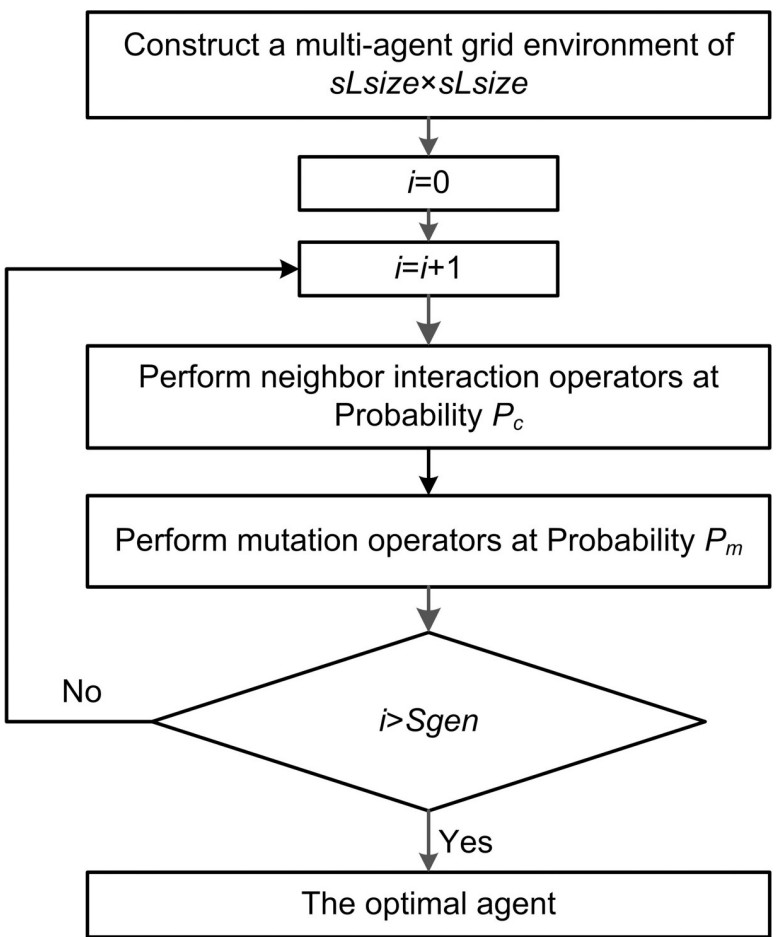

**Fig 12. Flow chart of mutation operator.**

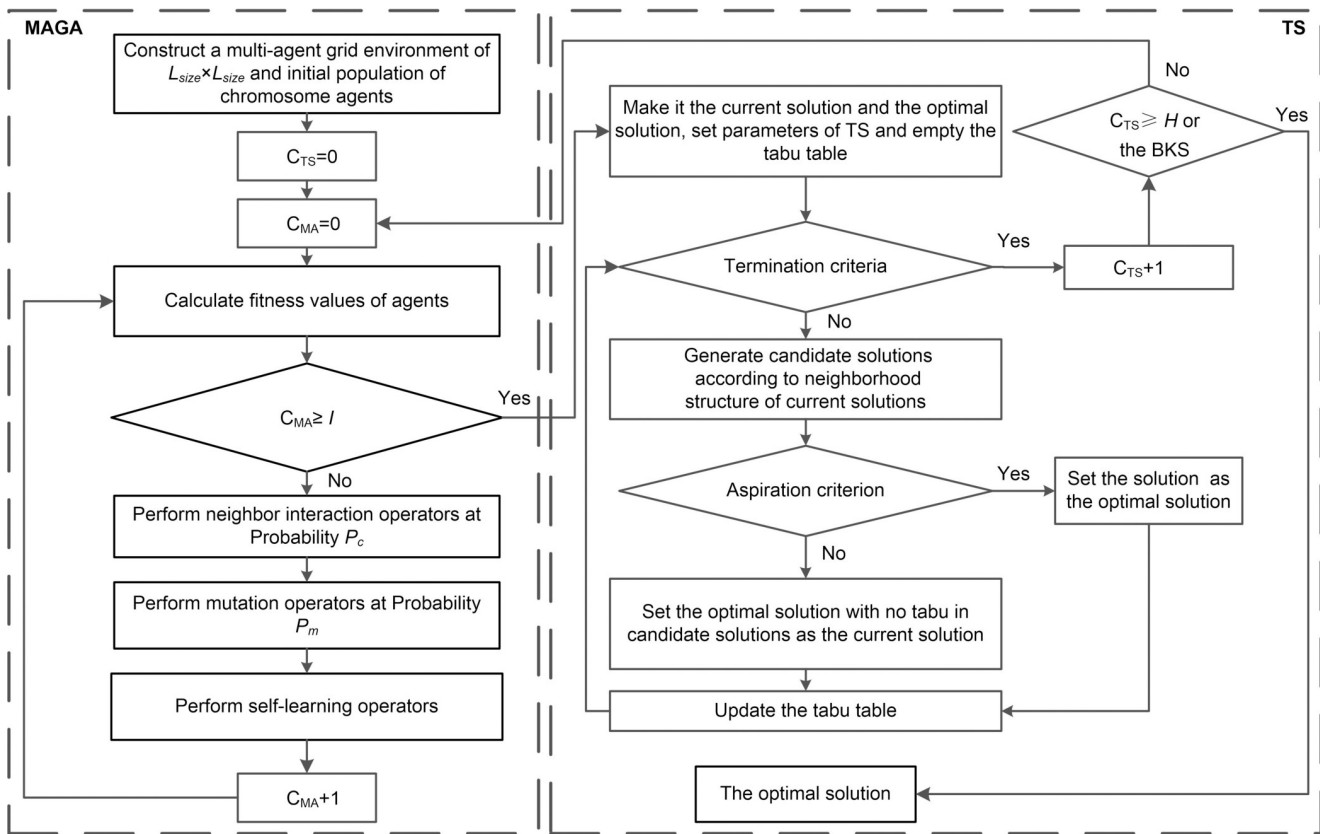

**Fig 13. Flow chart of MAGATS.**

(2) Obtain a set of candidate solutions to be selected according to neighborhood structure of current solutions (N7 neighborhood structure is adopted). Verify tabu of each candidate solution.

(3) Determine whether tabu solutions satisfy the aspiration criterion: if fitness of a tabu solution is higher than previous accepted solutions, then the solution will be released and will be set as the current solution and the optimal solution. If it does not, a solution with the highest fitness selected from the non-tabu solution set will be the current solution. Update the tabu table.

(4) Repeat steps (2), (3) until satisfying termination criteria. The termination criteria are to reach the maximum number of iterations or find the best known solution (BKS). Set the maximum number of iterations to 300.

(5) If the BKS is found, the algorithm is stopped. Otherwise, repeat steps (1)—(4) until the BKS is found or $C_{TS} \geq H$.

Key parameter settings of TS in MAGATS include selection of tabu objects and length setting of a tabu table.

(1) Selection of tabu objects. In section 2.2.3, neighborhood structures of JSSPs and judgement of key operations and exchangeable operation pairs were discussed. Therefore, operation pairs were marked as tabu objects. For example, a candidate solution is generated through exchanging operations $O_{12}$ and $O_{32}$ which are both processed on $M_1$. Then, exchange of this operation pair is added to the tabu table.

(2) Length setting of a tabu table. A tabu table can be regarded as a special queue with first-in, first-out characteristics. When a new tabu object is added, set its tabu length $L_{\text{list}}$. In other words, once entering in the tabu table, the tabu object can be dequeued until $L_{\text{list}}$ iterations of

TS (provided that aspiration criterion is not met). If tabu length of the tabu table is too small, it is easy to fall into local optimum. On the contrary, if tabu length is too large, too many constraints will be generated. An effective approach for JSSPs is to introduce a dynamic tabu table, which sets the tabu length $L_{list}$ to be randomly selected in $[L_{min}, L_{max}]$. After several tests, setting $L_{min} = [10+n/m]$ and $L_{max} = [1.8L_{min}]$ is better.

## 4 Results and analysis

The proposed algorithm MAGATS is implemented by Visual C++. The program running environment is a PC with a core i5 processor, 2.5GHz main frequency, and 8G memory. To test optimization performance of the algorithm, 43 instances are selected from the OR-library, which included FT06, FT10,FT20 contributed by Fisher and Thompson and LA01~LA40 contributed by Lawrence [28,29]. These instances are often used to test optimization performance of a new algorithm. Two comparative analysis are completed based on these instances.

### 4.1 Comparison of MAGATS with GA and MAGA

Parameters of GA, MAGA and MAGATS are set as follows.

GA: Size of initial population = 256, probability of crossover $Pc$ = 0.8, the number of crossover $\lambda$ = 50, probability of mutation $P_m$ = 0.1, the maximum number of iterations $I$ = 300. MAGA: $L_{size}$ = 16,$sL_{size}$ = 3,probability of neighbor interaction $Pc$ = 0.8, probability of replacement mechanism $P_o$ = 0.5,the number of crossover $\lambda$ = 50, probability of mutation $P_m$ = 0.1, the number of self-learning $S_{gen}$ = 50, the maximum number of iterations $I$ = 300; TS: the maximum number of cycles $H$ = 300.The remaining of relevant parameters refers to section 3. 11 instances of different sizes are selected to verify the improvement of MAGATS compared with GA and MAGA. The results are shown in Table 2. $C_{best}$ is the best solution every algorithm can find in 20 runs. $C_{aver}$ and $\sigma$ are the mean and standard deviation of all 20 solutions acquired by GA,MAGA and MAGATS.

As shown in Table 2 and Fig 14, compared with GA and MAGA. MAGATS finds the most BKSs and behaves better optimization performance and better stability.

### 4.2 Comparison between MAGATS and other algorithms

In this comparative analysis, all 43 instances are run 20 times respectively, and the obtained test results with MAGATS are compared with other four algorithms: NIMGA, aLSGA, WW,

**Table 2. Comparison of MAGATS with GA and MAGA.**

| Instances | Size | BKS | GA | | | MAGA | | | MAGATS | | |
|---|---|---|---|---|---|---|---|---|---|---|---|
| | | | $C_{best}$ | $C_{aver}$ | $\sigma$ | $C_{best}$ | $C_{aver}$ | $\sigma$ | $C_{best}$ | $C_{aver}$ | $\sigma$ |
| FT06 | 6×6 | 55 | 55 | 55 | 0 | 55 | 55 | 0 | 55 | 55 | 0 |
| FT10 | 10×10 | 930 | 930 | 949.7 | 15.05 | 930 | 939 | 7.42 | 930 | 930 | 0 |
| FT20 | 20×5 | 1165 | **1174** | 1193.55 | 13.67 | 1165 | 1173.40 | 4.76 | 1165 | 1165 | 0 |
| LA01 | 10×5 | 666 | 666 | 666 | 0 | 666 | 666 | 0 | 666 | 666 | 0 |
| LA06 | 15×5 | 926 | 926 | 926 | 0 | 926 | 926 | 0 | 926 | 926 | 0 |
| LA11 | 20×5 | 1222 | 1222 | 1222 | 0 | 1222 | 1222 | 0 | 1222 | 1222 | 0 |
| LA16 | 10×10 | 945 | **946** | 952.95 | 9.87 | **946** | 949 | 4.15 | 945 | 945.90 | 0.3 |
| LA24 | 15×10 | 935 | **947** | 969.60 | 12.56 | **943** | 958.35 | 9.10 | 935 | 943.25 | 3.16 |
| LA29 | 20×10 | 1152 | **1243** | 1267.05 | 16.93 | **1193** | 1214.4 | 11.28 | **1164** | 1174.15 | 8.05 |
| LA35 | 30×10 | 1888 | 1888 | 1888 | 0 | 1888 | 1888 | 0 | 1888 | 1888 | 0 |
| LA36 | 15×15 | 1268 | **1296** | 1326.55 | 12.75 | **1292** | 1299.27 | 6.33 | **1281** | 1288.70 | 4.87 |

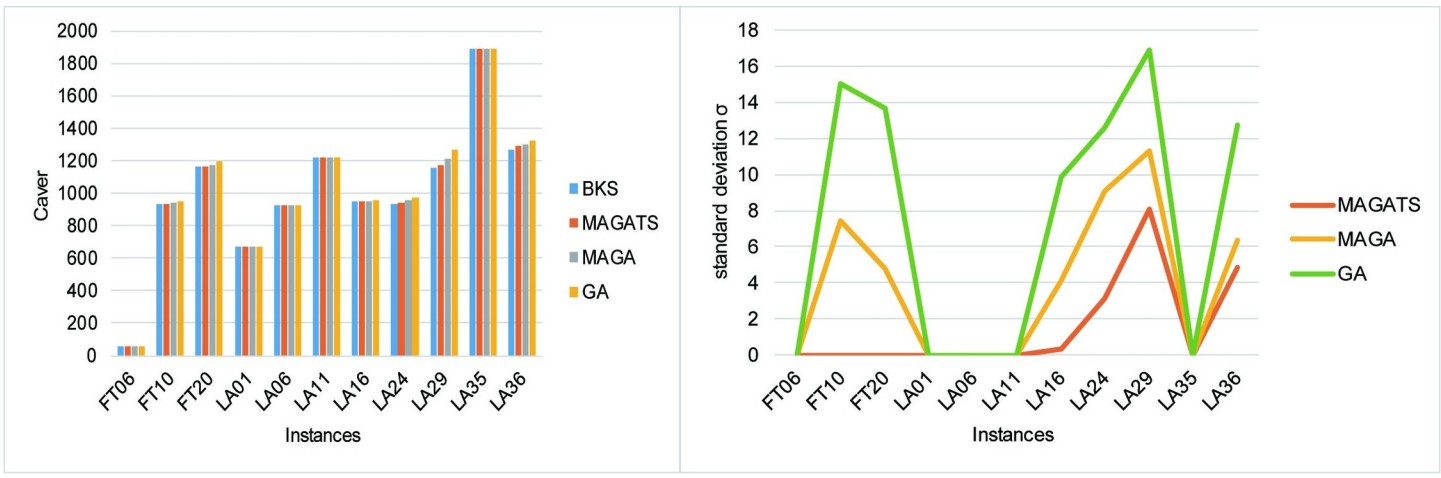

**Fig 14. Mean and standard deviation of GA, MAGA and MAGATS.**

TSSB [9,30–32], as shown in Table 3. Table 3 lists instance name, instance size (number of jobs × number of machines), the best known solution (BKS), the optimal solution (OS) obtained by MAGATS, RD (relative deviation between the optimal solution obtained by MAGATS and BKS) and the optimal solution obtained by other algorithms. RD can be calculated by Eq (4).

$$RD = (OS - BKS)/BKS \times 100\% \tag{4}$$

Based on the obtained RDs for each instance, ARD used to evaluate the optimization performance of each algorithm can be calculated by Eq (5).

$$ARD = \sum_{i=1}^{N} RD/N \tag{5}$$

where N is the number of instances tested by each algorithm. Contents of Table 4 include the number of instances solved/the number of instances tested (NIS/NIT), ARD of other algorithms(OA) and MAGATS and improvement. The column named improvement means the reduction of ARD obtained by MAGATS compared with OA. In other words, the more efficient algorithm can be identified.

Based on Tables 3 and 4, the new algorithm is analyzed.

(1) The proposed algorithm MAGATS finds 38 optimal solutions of 43 instances. To small instances FT06, FT10, FT20 and LA01~LA15, all five algorithms can find almost all the best known solutions. But to relatively large instances LA16~LA40 (25 instances), 20 best known solutions (80%) is found by MAGATS, which has better optimization performance than NIMGA(48%), aLSGA(45%), WW(64%) and TSSB (72%).

(2) Compared with other algorithms, ARD of MAGATS is the smallest. It indicates that the optimal solutions obtained by MAGATS is closest to the best known solutions or the minimum makespan. It is further shown that the new algorithm has better optimization performance. Compared with other algorithms, obtained solutions are of higher quality.

In order to visualize scheduling results of the JSSP optimized by MAGATS, an example LA39 is selected to display its optimal scheduling in a machine Gantt chart, as shown in Fig 15. Since $O_{ij}$ is easy to cause ambiguity, $(Ji,j)$ is used to represent the $j$th operation of $J_i$.

**Table 3. Comparison of optimization results of 43 instances.**

| Instances | Size | BKS | MAGATS | RD(%) | NIMGA | aLSGA | WW | TSSB |
|---|---|---|---|---|---|---|---|---|
| FT06 | 6×6 | 55 | 55 | 0 | 55 | 55 | 55 | 55 |
| FT10 | 10×10 | 930 | 930 | 0 | 930 | 930 | 930 | 930 |
| FT20 | 20×5 | 1165 | 1165 | 0 | **1173** | 1165 | 1165 | 1165 |
| LA01 | 10×5 | 666 | 666 | 0 | 666 | 666 | 666 | 666 |
| LA02 | 10×5 | 655 | 655 | 0 | 655 | 655 | 655 | 655 |
| LA03 | 10×5 | 597 | 597 | 0 | 597 | **606** | 597 | 597 |
| LA04 | 10×5 | 590 | 590 | 0 | 590 | **593** | 590 | 590 |
| LA05 | 10×5 | 593 | 593 | 0 | 593 | 593 | 593 | 593 |
| LA06 | 15×5 | 926 | 926 | 0 | 926 | 926 | 926 | 926 |
| LA07 | 15×5 | 890 | 890 | 0 | 890 | 890 | 890 | 890 |
| LA08 | 15×5 | 863 | 863 | 0 | 863 | 863 | 863 | 863 |
| LA09 | 15×5 | 951 | 951 | 0 | 951 | 951 | 951 | 951 |
| LA10 | 15×5 | 958 | 958 | 0 | 958 | 958 | 958 | 958 |
| LA11 | 20×5 | 1222 | 1222 | 0 | 1222 | 1222 | 1222 | 1222 |
| LA12 | 20×5 | 1039 | 1039 | 0 | 1039 | 1039 | 1039 | 1039 |
| LA13 | 20×5 | 1150 | 1150 | 0 | 1150 | 1150 | 1150 | 1150 |
| LA14 | 20×5 | 1292 | 1292 | 0 | 1292 | 1292 | 1292 | 1292 |
| LA15 | 20×5 | 1207 | 1207 | 0 | 1207 | 1207 | 1207 | 1207 |
| LA16 | 10×10 | 945 | 945 | 0 | 945 | **946** | 945 | 945 |
| LA17 | 10×10 | 784 | 784 | 0 | 784 | 784 | 784 | 784 |
| LA18 | 10×10 | 848 | 848 | 0 | 848 | 848 | 848 | 848 |
| LA19 | 10×10 | 842 | 842 | 0 | 842 | **852** | 842 | 842 |
| LA20 | 10×10 | 902 | **907** | 0.55 | **907** | **907** | **907** | 902 |
| LA21 | 15×10 | 1046 | 1046 | 0 | **1058** | **1068** | 1046 | 1046 |
| LA22 | 15×10 | 927 | 927 | 0 | **937** | **956** | **935** | 927 |
| LA23 | 15×10 | 1032 | 1032 | 0 | 1032 | 1032 | 1032 | 1032 |
| LA24 | 15×10 | 935 | 935 | 0 | **947** | **966** | **937** | **938** |
| LA25 | 15×10 | 977 | 977 | 0 | **989** | **1002** | 977 | **979** |
| LA26 | 20×10 | 1218 | 1218 | 0 | 1218 | **1223** | 1218 | 1218 |
| LA27 | 20×10 | 1235 | 1235 | 0 | **1269** | **1281** | **1236** | 1235 |
| LA28 | 20×10 | 1216 | 1216 | 0 | **1247** | **1245** | 1216 | 1216 |
| LA29 | 20×10 | 1152 | **1164** | 1.04 | **1241** | **1230** | **1160** | **1168** |
| LA30 | 20×10 | 1355 | 1355 | 0 | 1355 | 1355 | 1355 | 1355 |
| LA31 | 30×10 | 1784 | 1784 | 0 | 1784 | 1784 | 1784 | 1784 |
| LA32 | 30×10 | 1850 | 1850 | 0 | 1850 | 1850 | 1850 | 1850 |
| LA33 | 30×10 | 1719 | 1719 | 0 | 1719 | 1719 | 1719 | 1719 |
| LA34 | 30×10 | 1721 | 1721 | 0 | 1721 | 1721 | 1721 | 1721 |
| LA35 | 30×10 | 1888 | 1888 | 0 | 1888 | 1888 | 1888 | 1888 |
| LA36 | 15×15 | 1268 | **1281** | 1.03 | **1293** | - | **1279** | 1268 |
| LA37 | 15×15 | 1397 | 1397 | 0 | **1432** | - | **1407** | **1411** |
| LA38 | 15×15 | 1196 | **1198** | 0.17 | **1222** | - | 1196 | **1201** |
| LA39 | 15×15 | 1233 | 1233 | 0.00 | **1251** | - | **1242** | **1240** |
| LA40 | 15×15 | 1222 | **1228** | 0.49 | **1246** | - | **1229** | **1233** |

## 5 Conclusion

Aiming at NP characteristics of JSSPs and minimizing makespan, a MAGATS is proposed in this paper. The new algorithm is applied to test 43 benchmark instances. Compared with four

**Table 4. Improvement of MAGATS compared with OA.**

| Algorithm | NIS/NIT | ARD | | Improvement |
|---|---|---|---|---|
| | | OA(%) | MAGATS(%) | |
| NIMGA | 29/43 | 0.68 | 0.08 | 0.60 |
| aLSGA | 25/38 | 0.74 | 0.04 | 0.70 |
| WW | 34/43 | 0.12 | 0.08 | 0.04 |
| TSSB | 36/43 | 0.11 | 0.08 | 0.03 |

other algorithms, optimization performance of it is analyzed based on the obtained test results. Analysis results show that the proposed algorithm MAGATS has high optimization performance and practical value in the field of JSSPs.

Highlights of this paper can be concluded into the following 3 aspects.

(1) A neighbor interaction operator based on a POX operator is designed. Under the algorithm framework of MAGA, each agent can only interact with neighbors. A replacement mechanism is adopted to retain good chromosomes. The algorithm's global search performance is enhanced to some extent, thus achieving optimization function of the algorithm.

(2) A mutation operator based on neighborhood structure and a self-learning operator are designed. By introducing N7 neighborhood structure into the design of mutation operators, a wider solution space can be explored, thereby obtaining a higher quality feasible scheduling. Centralized search ability of MAGATS can be further enhanced by the self-learning operator.

(3) A MAGATS is proposed. Combined with excellent global search performance of TS, MAGA is integrated with TS to further enhance optimization performance of the algorithm, avoiding premature and falling into local optimal.

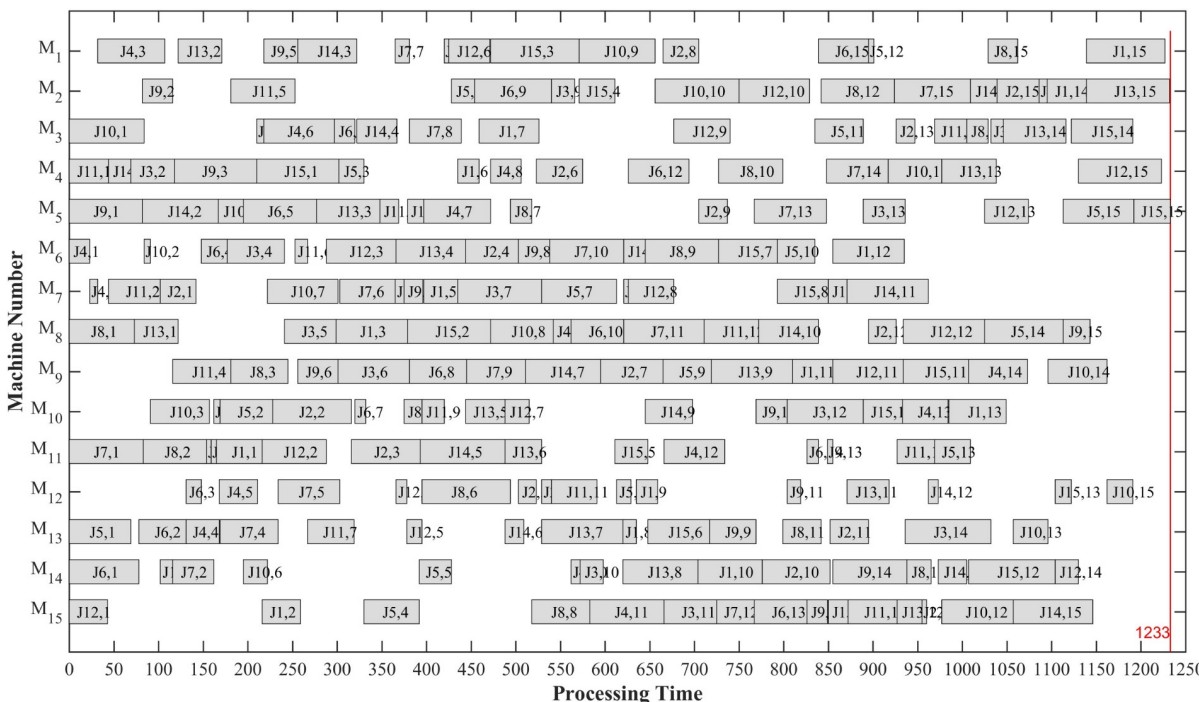

**Fig 15. Machine Gantt chart of instance LA39.**

The proposed algorithm MAGATS is only used to solve JSSPs with a goal of minimizing makespan. Influences of neighbor environment of chromosome agents on optimization performance of the algorithm need further research. In the future research work, feasibility of this algorithm in multi-objective job shop scheduling and flexible job shop scheduling will also be studied.

## Supporting information

**S1 File. Meta-data of 43 instances.**
(DOCX)

## Acknowledgments

I would like to express my sincere gratitude to my teammates, Wenguang Zuo, Caiyu Zhen and Miao Chen. During the writing and revision of the paper, they provided great help for me.

## Author Contributions

**Conceptualization:** Chong Peng.

**Data curation:** Guanglin Wu.

**Methodology:** Guanglin Wu.

**Software:** Guanglin Wu.

**Supervision:** Chong Peng.

**Validation:** Guanglin Wu.

**Writing – original draft:** Guanglin Wu.

**Writing – review & editing:** Guanglin Wu, T. Warren Liao, Hedong Wang.

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
