## [Decision Letter · Decision Letter 0]

11 Sep 2019

[EXSCINDED]

PONE-D-19-21845

Research on Multi-agent Genetic Algorithm Based on Tabu Search for Job Shop Scheduling Problem

PLOS ONE

Dear Professor Peng,

Thank you for submitting your manuscript to PLOS ONE. After careful consideration, we feel that it has merit but does not fully meet PLOS ONE’s publication criteria as it currently stands. Therefore, we invite you to submit a revised version of the manuscript that addresses the points raised during the review process.

Please check the reference citation format. For example, "Garey M and Sethi J proved that JSSP had non-deterministic polynomial (NP) characteristics [1]. ". "Garey and Sethi proved that JSSP had non-deterministic polynomial (NP) characteristics [1]. ".

We would appreciate receiving your revised manuscript by Oct 26 2019 11:59PM. To enhance the reproducibility of your results, we recommend that if applicable you deposit your laboratory protocols in protocols.io, where a protocol can be assigned its own identifier (DOI) such that it can be cited independently in the future. For instructions see: http://journals.plos.org/plosone/s/submission-guidelines#loc-laboratory-protocols

We look forward to receiving your revised manuscript.

Kind regards,

Feng Chen

Academic Editor

PLOS ONE

Journal Requirements:

Reviewers' comments:

Reviewer's Responses to Questions

**Comments to the Author**

1. Is the manuscript technically sound, and do the data support the conclusions?

Reviewer #1: Yes

Reviewer #2: Yes

2. Has the statistical analysis been performed appropriately and rigorously? 

Reviewer #1: Yes

Reviewer #2: Yes

3. Have the authors made all data underlying the findings in their manuscript fully available?

Reviewer #1: Yes

Reviewer #2: Yes

4. Is the manuscript presented in an intelligible fashion and written in standard English?

Reviewer #1: Yes

Reviewer #2: Yes

5. Review Comments to the Author

Reviewer #1: The manuscript entitled “Research on multi-1 agent genetic algorithm based on tabu search for job shop scheduling problem” by Chong Peng et al. proposes a multi-agent genetic algorithm based on tabu search to solve job shop scheduling problem under makespan constraints.

1. The authors presents an algorithm combining tabu search with a MAGA. With benchmark instances, the algorithm proves its improvement in effectiveness.

2. The statistical analysis has been performed with 43 benchmark instances often used to test optimization performance. The results reported a better optimization performance. The authors conclusion sounds credible.

3. The language of the manuscript looks good.

Reviewer #2: The solution to job shop scheduling problem is of great significance for improving resource utilization and production efficiency of enterprises. In this paper, in view of its non-deterministic polynomial properties, a multi-agent genetic algorithm based on tabu search (MAGATS) was proposed to solve job shop scheduling problem under makespan constraints. Firstly, a multi-agent genetic algorithm (MAGA) was proposed. The paper is well organized and the methods proposed by the authors were verified by the given examples. I can recommend it to be accepted.

6. PLOS authors have the option to publish the peer review history of their article (what does this mean?). If published, this will include your full peer review and any attached files.

Reviewer #1: Yes: Pengzhong, LI

Reviewer #2: No

---

## [Author Response · Author response to Decision Letter 0]

14 Sep 2019

Dear Reviewers and Editors,

Thank you very much for your useful comments and suggestions on the content and language of our manuscript. Your comments are very valuable and helpful for revising and improving our work. The detailed corrections are written in the file "Response to Reviewers". I copy the contents here.

Reviewer #1：

1) Combining multi-agent synergy theories and genetic algorithm, a multi-agent genetic algorithm (MAGA) is firstly proposed. Furthermore, a multi-agent genetic algorithm based on tabu search (MAGATS) is proposed. This paper only discusses the improvement of optimization performance of the MAGATS from the theoretical level, but lacks case analysis. It is better to compare MAGATS with MAGA and GA to show the difference multi-agent and TS make.

Thanks for the comments.

Your comments are very helpful to further improve the logical rigor of our paper. According to your comments, in Section 4.1 of the revised manuscript, we use 11 instances of different sizes to test the performance of MAGATS, GA and MAGA. By calculating the optimal value, average value and variance of the optimization results, the optimization quality and stability of the MAGATS compared with the GA and MAGA is proved. Detailed changes are in the revised manuscript with track changes.

Please check.

2) The structure of the paper can be further improved. For example, Section 2 is too small compared with other major sections.

Thanks for the comments.

The contents described in Section 2 of the manuscript is mainly about the model of job shop scheduling. The achievement of MAGA and MAGATS is introduced successively in Section 3. Compared with Section 3, the information provided is relatively litter in Section 2. According to your comments, Section 3 is split two sections, that is, “The achievement of MAGA” and “The achievement of MAGATS”. Section “Model of JSSP” and Section “The achievement of MAGA” are merged into one new Section “The achievement of MAGA”. Detailed changes are in the revised manuscript with track changes.

Please check.

3) Please check the reference citation format. For example, "Garey M and Sethi J proved that JSSP had non-deterministic polynomial (NP) characteristics [1]. ". "Garey and Sethi proved that JSSP had non-deterministic polynomial (NP) characteristics [1]. "

Thanks for the comments.

Per the Reviewer’s suggestion, we have made corresponding corrections and check the references citation format carefully. Detailed changes are in the revised manuscript with track changes.

Please check.

4) Define abbreviations upon first appearance in the text. Please check the abbreviations in the paper. For example, the abbreviations “GA” of “genetic algorithm” should appear in the abstract.

Thanks for the comments.

Per the Reviewer’s suggestion, we have made corresponding corrections and check the abbreviations carefully. Detailed changes are in the revised manuscript with track changes.

Please check.

5) The article is too long and should be condensed. The contents of the paper should mainly reflect its own important research contents.

Thanks for the comments.

Per the Reviewer’s suggestion, we cut down the contents of three places in the paper.

Firstly, in Section 1, the contents about exact methods for solving JSSPs is cut down. The researches about mathematical programming method, branch and bound method are not relevant to our research.

Secondly, in Section 3.1.1, Table 1 is deleted. The table is a short review about advantages and disadvantages encoding methods of GA used for JSSP. The main purpose of this paper is to choose the most suitable operation-based encoding method by comparing various encoding methods. The table is not necessary in this paper. Pointing out the advantages of the operation-based encoding method is enough. 

Finally, Fig 5 in the manuscript is about the semi-active scheduling solution of 4×4 JSSP in Table 2. Its original purpose is to prove that active scheduling is superior to semi-active scheduling, which is an accepted fact. It is also unnecessary in the paper. Detailed changes are in the revised manuscript with track changes.

Please check.

Reviewer #2：

1) It is better to show the mean and standard deviation of 20-run results for all algorithms, which helps to see whether the differences are statistically significant or not.

Thanks for the comments.

Table 3 in Section 4 contains the optimization results of 5 algorithms for 43 instances. For other algorithms other than MAGATS in the table, the corresponding paper only provides the optimization results for various instances, but does not provide the mean and standard deviation of the algorithm. Therefore, considering the time cost, it is difficult to implement. However, in the new Section 4.1, the mean and standard deviation of GA, MAGA and MAGATS are calculated, which is used to complete the comparative analysis of three algorithms. Detailed changes are in the revised manuscript with track changes.

Please check.

2) In section 3.2. Fig 14 introduces the flow chart of MAGATS. But the parameters setting and specific steps of TS are introduced below the figure. There is some confusion in logic, so it is better to reorganize it.

Thanks for the comments.

MAGATS is a combination of MAGA and TS. Because the flow chart of MAGA has been introduced in Section 3.1.1, this section focuses on the remaining TS in MAGATS. Therefore, below the figure, we only introduce the parameters setting and specific steps of TS. To avoid logical misunderstanding of the content of the paper，we have made some revisions in the revised manuscript with track changes. “Specific steps of TS are summarized as follows.” is modified to “Specific steps of MAGATS are summarized as follows.” The corresponding contents of some steps are also modified to some extent. Detailed changes are in the revised manuscript with track changes.

Please check.

3) The tenses in the article are confused, the simple present tense and the simple past tense coexist. The unified tense is helpful to understand the article.

Thanks for the comments.

Per the Reviewer’s suggestion, the tense in our paper is modified to the simple past tense. The main changes are in the abstract and the conclusions. Detailed changes are in the revised manuscript with track changes.

Please check.

4) The variables in the text should be italicized, such as variable “λ” in Row 201 and Row 203.

Thanks for the comments.

Per the Reviewer’s suggestion, we have made corresponding corrections. Detailed changes are in the revised manuscript with track changes.

Please check.

---

## [Editor Report · Decision Letter 1]

17 Sep 2019

Research on Multi-agent Genetic Algorithm Based on Tabu Search for Job Shop Scheduling Problem

PONE-D-19-21845R1

Dear Dr. Peng,

We are pleased to inform you that your manuscript has been judged scientifically suitable for publication and will be formally accepted for publication once it complies with all outstanding technical requirements.

With kind regards,

Feng Chen

Academic Editor

PLOS ONE

Journal Requirements: 

We find that there are remaining concerns related to the English language and syntax. For instance, "job shop scheduling problem" should be changed to "the job shop scheduling problem" when it is written out in the title, abstract, and introduction. (If it is plural ["job shop scheduling problems"], "the" is not needed.) Please address this as part of of your technical revisions. Note that PLOS ONE does not provide in-house copyediting. Thus, we would recommend having the full manuscript checked again by a copyeditor prior to publication.
---

## [Editor Report · Acceptance letter]

20 Sep 2019

PONE-D-19-21845R1 

Research on multi-agent genetic algorithm based on tabu search for the job shop scheduling problem

Dear Dr. Peng:

I am pleased to inform you that your manuscript has been deemed suitable for publication in PLOS ONE. Congratulations! Your manuscript is now with our production department. 

With kind regards,

on behalf of

Dr. Feng Chen 

Academic Editor

PLOS ONE